# Periorbital Nociception in a Progressive Multiple Sclerosis Mouse Model Is Dependent on TRPA1 Channel Activation

**DOI:** 10.3390/ph14080831

**Published:** 2021-08-23

**Authors:** Diéssica Padilha Dalenogare, Camila Ritter, Fernando Roberto Antunes Bellinaso, Sabrina Qader Kudsi, Gabriele Cheiran Pereira, Maria Fernanda Pessano Fialho, Débora Denardin Lückemeyer, Caren Tatiane de David Antoniazzi, Lorenzo Landini, Juliano Ferreira, Guilherme Vargas Bochi, Sara Marchesan Oliveira, Francesco De Logu, Romina Nassini, Pierangelo Geppetti, Gabriela Trevisan

**Affiliations:** 1Graduated Program in Pharmacology, Federal University of Santa Maria (UFSM), Santa Maria 97105-900, RS, Brazil; diessica_dalenogare@hotmail.com (D.P.D.); milla_ritter@hotmail.com (C.R.); fernando.bellinaso@gmail.com (F.R.A.B.); sabrina.qader@acad.ufsm.br (S.Q.K.); gabrielecheiran@gmail.com (G.C.P.); caren.antoniazzi@dcu.ie (C.T.d.D.A.); guilherme.bochi@ufsm.br (G.V.B.); 2Graduated Program in Biological Sciences: Toxicological Biochemistry, Federal University of Santa Maria (UFSM), Santa Maria 97105-900, RS, Brazil; mariafpessano@outlook.com (M.F.P.F.); saramarchesan@hotmail.com (S.M.O.); 3Graduated Program in Pharmacology, Federal University of Santa Catarina (UFSC), Florianópolis 88040-900, SC, Brazil; debora_dl@hotmail.com (D.D.L.); ferreiraj99@gmail.com (J.F.); 4Department of Health Science, Clinical Pharmacology and Oncology, University of Florence, 50139 Florence, FI, Italy; l.landini@unifi.it (L.L.); francesco.delogu@unifi.it (F.D.L.); pierangelo.geppetti@unifi.it (P.G.)

**Keywords:** metamizole, anti-oxidants, calcitonin gene-related peptide, sumatriptan, migraine, headache

## Abstract

Headaches are frequently described in progressive multiple sclerosis (PMS) patients, but their mechanism remains unknown. Transient receptor potential ankyrin 1 (TRPA1) was involved in neuropathic nociception in a model of PMS induced by experimental autoimmune encephalomyelitis (PMS-EAE), and TRPA1 activation causes periorbital and facial nociception. Thus, our purpose was to observe the development of periorbital mechanical allodynia (PMA) in a PMS-EAE model and evaluate the role of TRPA1 in periorbital nociception. Female PMS-EAE mice elicited PMA from day 7 to 14 days after induction. The antimigraine agents olcegepant and sumatriptan were able to reduce PMA. The PMA was diminished by the TRPA1 antagonists HC-030031, A-967079, metamizole and propyphenazone and was absent in TRPA1-deficient mice. Enhanced levels of TRPA1 endogenous agonists and NADPH oxidase activity were detected in the trigeminal ganglion of PMS-EAE mice. The administration of the anti-oxidants apocynin (an NADPH oxidase inhibitor) or alpha-lipoic acid (a sequestrant of reactive oxygen species), resulted in PMA reduction. These results suggest that generation of TRPA1 endogenous agonists in the PMS-EAE mouse model may sensitise TRPA1 in trigeminal nociceptors to elicit PMA. Thus, this ion channel could be a potential therapeutic target for the treatment of headache in PMS patients.

## 1. Introduction

The transient receptor potential ankyrin 1 (TRPA1) is widely expressed in mammalian tissues, including the sensory neurons of the trigeminal ganglion (TG) [1,2,3]. TRPA1 is involved in acute and chronic pain generation, and its activation is mediated by several exogenous and endogenous agonists [1,4,5,6]. The main TRPA1 endogenous agonists produced by the inflammatory process and oxidative stress are hydrogen peroxide (H_2_O_2_) and 4-hydroxynonenal (4-HNE) [1,7]. These agonists present increased levels in different mouse and rat models of periorbital and facial nociception, such as migraine-like behaviour induced by glyceryl trinitrate (GNT) [8] and trigeminal neuropathic pain caused by constriction of the infraorbital nerve [9]. Additionally, dural injection of TRPA1 agonists causes migraine-related periorbital mechanical allodynia (PMA) [10]. For these reasons, TRPA1 has been studied as a valuable therapeutic target for headache and migraine treatment [11].

Moreover, TRPA1 activation leads to calcitonin gene-related peptide (CGRP) release [12,13,14,15,16], an endogenous compound related to headache development [1,10,12]. CGRP release leads to local and systemic vasodilation, one of the primary known headache- and migraine-related mechanisms [13,17,18]. Gepants (CGRP receptor antagonists) are the newest compounds to be used clinically for acute migraine treatment [18,19]. Triptans, serotonin (5HT)IB/D-like receptor agonists, are also used for migraine crisis treatment [20,21]. In addition, sumatriptan (a triptan) [10,22,23] and olcegepant (a gepant) [8,24,25] are extensively used, as positive control drugs, during the optimisation of different migraine-like mouse models. On the other hand, gepants and triptans generate diverse adverse effects, such as cardiovascular and central nervous system alterations [19,26,27,28,29]. Thus, the development of new drugs for headache and migraine treatment is demanded.

In these circumstances, headache can be present in several pathologies, including progressive multiple sclerosis (PMS), a neuroinflammatory autoimmune disease that presents neuropathic pain and headache as leading painful symptoms [30,31,32,33]. The headache symptom is observed in the early stages of MS [34,35]. Based on the International Classification of Headache Disorders (ICHD-2), a previous study demonstrated that the rate of headaches in MS patients was 4%–61.8% [36], migraine headache and tension-type headache being the main headache types described [37]. However, the treatment of headache and migraine in MS is still not adequate, indicating the necessity to study new mechanisms and develop models for this pathology.

Recently, the development was described of trigeminal neuropathic pain behaviours and facial hypersensitivity in the whisker pad area in a progressive multiple sclerosis induced by experimental autoimmune encephalomyelitis (PMS-EAE) model in mice [38,39]. When the whisker pad area is stimulated the lesion or inflammation process in the trigeminal nerve is usually analysed [40,41]. However, until now no study has described the development of PMA in this model, which was demonstrated as a headache- and migraine-related area for the development of mechanical allodynia [8,10,23,24,42].

Previously, our group described the relation between the development of neuropathic pain-like behaviours and TRPA1 activation induced by two different MS mouse models: the PMS-EAE and the relapsing-remitting multiple sclerosis (RR-EAE) models [43,44]. Thus, our main purpose was to optimise the development of PMA in a PMS-EAE model and detect the role of TRPA1 in this nociceptive behaviour.

## 2. Results

### 2.1. PMS-EAE Caused PMA Development, but Trpa1^−/−^ Mice Induced to PMS-EAE Did Not Develop PMA and Treatment with Antimigraine Agents Reduced Periorbital Nociception

PMS-EAE-induced mice showed a reduction in periorbital mechanical threshold from day 7 to 14 post-induction (p.i.; Figure 1A). MOG_35–55_ and CFA administration elicited an increase in clinical scores that started at day 13 p.i. (Appendix A), indicating the onset of PMS-EAE, as previously reported [44]. Nevertheless, no changes in locomotor activity (Appendix A) or body weight (Appendix A) between the PMS-EAE and the control groups were detected during the 14 days of observation. However, as the animals after 14 days showed motor impairment and weight loss, we selected this time point to perform our study, as our previous publication that used this PMA-EAE model [44]. In addition, the periorbital mechanical threshold of PMS-EAE mice was 0.0126 ± 0.0002 g; for control it was 0.4273 ± 0.2177 g 14 days p.i., which shows that the PMS-EAE-induced mice had a very low threshold considering that the lowest von Frey filament used was 0.008 g.

Moreover, we showed that *Trpa1^−/−^* mice did not develop PMA after PMS-EAE induction when compared to the *Trpa1^+/+^* group (Figure 2B). The systemic administration of sumatriptan (0.6 mg/kg, i.g.) at day 14 p.i. was able to attenuate PMA at 1, 2, and 3 h (h) after its administration, with maximum inhibition (I_max_) of 100% at 2 h post-administration (Figure 1C). A similar effect was detected after the CGRP antagonist (olcegepant, 1 mg/kg, i.p.) administration 14 days p.i., with antiallodynic effect occurrence at 1 and 2 h after treatment, and with an I_max_ of 96 ± 6%, 2h after its administration (Figure 1D).

### 2.2. TRPA1 Antagonists Attenuated PMA in PMS-EAE-Induced Mice

The administration of TRPA1 antagonists, HC-030031 (300 mg/kg, i.g.), A-967079 (100 mg/kg, i.g.), metamizole (100 mg/kg, i.g.), and propyphenazone (100 mg/kg, i.g.), resulted in antinociceptive effects on the PMA caused by PMS-EAE (Figure 2). TRPA1 selective antagonists HC-030031 (Figure 2A) and A-967079 (Figure 2B) reduced PMA from 1 and 2 h after i.g. administration. The I_max_ values obtained for HC-030031 and A-967079 were 92 ± 16% and 100%, respectively, 1h after treatment. Moreover, the antiallodynic effect of metamizole (Figure 2C) and propyphenazone (Figure 2D) was seen from 1 to 2 h after administration. For metamizole and propyphenazone, the I_max_ values were 69 ± 25% and 80 ± 13%, respectively, at 1 h after treatment.

### 2.3. TRPA1 Endogenous Agonist Levels and NADPH Oxidase Activity Were Increased in PMS-EAE Induced Mice

The PMS-EAE induction was able to increase oxidative markers in the trigeminal ganglion (Figure 3A–C) of induced mice. The levels of TRPA1 agonists 4-HNE (Figure 3A) and H_2_O_2_ (Figure 3B) were increased, as well as NADPH oxidase (Figure 3C) activity in the trigeminal ganglion of the PMS-EAE induced mice when compared to the control group 14 days p.i.

### 2.4. The Intragastric Treatment of α-Lipoic Acid and Apocynin Attenuated PMA in the PMS-EAE in Induced Mice

Administration of α-lipoic acid (100 mg/kg, i.g.) (Figure 4A) and apocynin (100 mg/kg, i.g.) (Figure 4B) demonstrated an antinociceptive effect in the PMS-EAE model from 1 to 2 h after intragastric administration. The I_max_ value was 100% for α-lipoic acid and 78 ± 23% for apocynin at 1 h after their administration.

## 3. Discussion

According to the literature, headache and neuropathic pain are the most prevalent pain symptoms in MS patients [45]. The headache in MS occurs during any stage of the disease [46]. In addition, headache symptoms were more prevalent in women: this symptom presents a ratio of three females affected for every male in MS [34,37,46]. However, headache treatment in MS patients is the same as that prescribed for the general population, and there is no difference in the management of this symptom between the sexes [47]. Until now, no study has evaluated the PMA response in the PMS-EAE model or evaluated the role of TRPA1 in this nociceptive response. Considering that TRPA1 is involved in headache and migraine mechanisms [8,9,10], we aimed to investigate the development of PMA in a PMS-EAE mouse model and we also intended to evaluate the involvement of TRPA1.

In addition to that, the frequency of headache symptoms leads to medicine overuse during the acute crisis leading to a vicious cycle of medication intake [48]. Thus, it is relevant to characterise the development of PMA in this model of PMS-EAE to evaluate novel mechanisms involved in this nociceptive response. In this study, we observed that the withdrawal threshold decreased in the periorbital area 7 days after the induction and persisted until 14 days post-induction. Previously, we showed the development of neuropathic pain-like behaviours, without motor impairment, with a nociception peak 14 days p.i. of PMS-EAE for mechanical and cold allodynia assessed in the hind paw [44]. We have not evaluated the PMA after 14 days p.i. because we aimed to detect this nociceptive response in a stage of PMS-EAE where animals did not lose weight or show debilitating clinical signs. We did not demonstrate non-evoked nociceptive tests in this present study; however, it is an important suggestion for future investigations. Recently, it was detected that after the induction of PMS-EAE there is an enhancement of mouse grimace scale scores and mechanical allodynia detected in the whisker pad [39]. Moreover, other measures could be used to detect spontaneous pain in models of facial nociception, such as conditioned place preference tests [49].

One of the leading drug classes used during acute migraine crisis treatment is the triptans. The triptans are serotonin (5HT) IB/D-like receptor agonists that produce high constriction of the cranial dural blood vessels [41,42]. Our results demonstrated that the sumatriptan administration caused an antiallodynic effect in the PMS-EAE induced mice. The peak concentration of sumatriptan is 45 min after oral administration in humans and shows a half-life of around 2 h [50,51]. The acute antimigraine effect in humans is short (around 4 h after the intake) [51]. Our data are also in accordance with the umbellulone migraine-like model that detected the antinociceptive effect of sumatriptan, using a measure of PMA. In this model also the antinociceptive effect of this compound also lasted approximately 4 h after i.g. treatment [52].

Furthermore, the CGRP receptor and its antagonists emerged as a new targets for migraine treatment in the clinical setting [53,54,55]. CGRP is a potent vasodilator in vascular beds and is related to nociceptive transmission in the trigeminal ganglion add [55]. In addition, various studies have shown that TRPA1 activation by its agonists leads to CGRP release [12,13,14,15,16]. The administration of olcegepant was able to attenuate the PMA observed in this PMS-EAE model. Olcegepant was the first gepant developed but it was discontinued because of difficulties in developing an oral formulation add [55] and is currently used as a CGRP antagonist for pathophysiological investigations into headache- and migraine-like rodent models by i.p. injection. Olcegepant shows a half-life of 2.5 h in humans after intravenous administration [56]. We detected an effect of this compound until 2 h after i.p. administration, but other studies described a different time curve effect with a detectable antinociceptive effect until 3–4 h after i.p. injection of olcegepant [24,52]. However, no previous no study has described the efficacy of this compound in a model of MS. With this in view, we characterised the development of PMA in a PMS-EAE model in mice, and this nociceptive behaviour was reduced by the injection of antimigraine agents. Unfortunately, we could not distinguish whether the PMA observed in this model was caused by headache, but most of the studies in this area used the measure of PMA as an indication of headache in animal models [8,24,42,57].

In addition to triptans and CGRP antagonists, common painkillers are used to treat primary headache symptoms, such as metamizole (dipyrone) or propyphenazone, which acts by cyclo-oxygenase inhibition without anti-inflammatory effects [58]. Our findings showed a reduction in PMA after metamizole (dipyrone) or propyphenazone injection when compared to their vehicle-induced groups, also for 1 and 2 h. We tested these two recently discovered TRPA1 non-selective antagonists [58] because of their use as analgesics in different forms of pain in humans [59,60,61,62]. It has already been described that metamizole and propyphenazone show antinociceptive effects until 1 h of administration [58]. The main metabolite of metamizole, the 4-methyl-amino-antipyrine (MMA) moiety, has a half-time of 2.6 to 3.5 h in humans after oral administration [63], and for propyphenazone approximately 2.8 h is described for oral administration [64]. In addition, we showed the periorbital antiallodynic effects of HC-030031 and A-967079 in a PMS-EAE model for 1 and 2 h. The TRPA1 antagonism has been reported to attenuate the trigeminal pain and migraine-like behaviours in different rodent models of facial pain with a similar duration of antinociceptive effect [9,12,65]. HC-030031 has a short half-life (32 min) in rats, and A-967079 has a distribution half-life of 1.8 h in rats [66]. HC-030031 and A-967079, when administered in mice in the doses used in this study, show antinociceptive effects for 1–2 h in other pain models [9,67]. Additionally, these TRPA1 antagonists (HC-030031, A-967079, metamizole, and propyphenazone) have antinociceptive effects during the mechanical and cold allodynia measurement in a neuropathic pain behaviour induced by PMS-EAE after 1–2 h [44].

Previously, in an MS mouse model induced by cuprizone, the role of TRPA1 in the demyelination process was demonstrated [68]. The genetic deletion of TRPA1 channels was able to reduce the myelin degeneration, and consequently, attenuated the behaviour impairment caused by cuprizone administration in mice [68,69]. Moreover, the genetic deletion of TRPA1 impairs the development of facial pain in different mouse models [8,9,11]. In agreement with these previous results, we also evaluated the development of PMA in PMS-EAE induced TRPA1 wild-type mice and observed the prevention of PMA occurrence in the TRPA1 knockout mice. Thus, using genetic and pharmacological tools we detected the role of TRPA1 in the PMA observed in a model of PMS-EAE in mice.

Diverse factors can activate the TRPA1 channel, such as exogenous and endogenous compounds [1]. Many natural compounds work as a TRPA1 agonists, such as cinnamaldehyde (found in cinnamon oil) [70], allyl or benzyl isothiocyanate (present in mustard oil and wasabi) [71] and allicin (found in garlic) [72]. Furthermore, some environmental irritants can activate TRPA1 channels, such as acrolein [14], tear gas [73], and aldehydes (present in cigarette smoke) [74,75]. In addition, several by-products of cellular respiration can also activate the TRPA1 channels such as reactive nitrogen, oxygen, and carbonyl species [1]. Reactive oxygen species (ROS), such as H_2_O_2_, singlet oxygen, ozone, and other organic oxides, are generated by the NADPH oxidase enzyme family [76]. Additionally, during the oxidative stress process, reactive nitrogen species (RNS), such as nitric oxide and peroxynitrite are produced by nitric oxidase synthases [76]. Enhancement of ROS and RNS production leads to the membrane peroxidation and generates reactive carbonyl species, such as 4-HNE and methylglyoxal [76]. Under pathological conditions and oxidative stress processes, these by-products could activate the TRPA1 channels, causing the release of neurogenic peptides, substance *p* and CGRP, and generate cold and mechanical hypersensitivity [7,15,16,66,77,78].

In this study we have evaluated the involvement of H_2_O_2_ and 4-HNE in PMS-EAE induction of PMA, but other endogenous agonists of TRPA1, such as acrolein and methylglyoxal, are also relevant for MS pathophysiology [79,80,81,82,83]. In EAE-induced mice, it the increase in acrolein protein adduct levels was demonstrated [84] and an increase in 3-hydroxypropylmercapturic, an acrolein glutathione metabolite, was detected in MS patients when compared to the control group [80,82]. In addition, the use of hydralazine, an acrolein scavenger, demonstrated an attenuation of the symptom severity and diminished disease EAE mice and could be a promising treatment mechanism in MS patients [83]. It was also reported described that methylglyoxal-derived advanced glycation end products are found in MS lesions in patients and may be correlated with MS pathology [79]. Methylglyoxal [85] and acrolein [86] mediate nociception in pain models, but no studies have described their role in MS pain or nociception.

The development of MS is characterised by an extensive inflammatory process and oxidative stress in the CNS, and the lesions lead to a meningeal irritation, as well as the production of reactive species and activation of trigeminal primary afferent nociceptive neurons [87,88]. Different facial pain models show alterations in the oxidative stress parameter and TRPA1 agonist levels [8,9]. Additionally, the PMS-EAE model showed an increase in 4-HNE levels and NADPH oxidase activity in the spinal cord [44]. Moreover, it has been reported that NADPH oxidase is involved in MS physiopathology by leading the generation of reactive species, and its increase has been shown in the brain, and spinal cord samples of PMS-EAE-induced mice [89].

Furthermore, we observed increases in H_2_O_2_ and 4-HNE levels and NADPH oxidase activity in trigeminal ganglion samples after PMS-EAE induction. We chose to detect the levels of TRPA1 agonists in the trigeminal ganglion because different mouse models have demonstrated a relation between facial allodynia and inflammatory/oxidative stress processes in the trigeminal ganglia [8,38,90,91]. Moreover, in a previous study using the same PMS-EAE mouse model, the development of facial hypersensitivity was observed when air puff was applied to the whisker pad area. Additionally, T-cell infiltration and glial activation were shown the in trigeminal primary afferents [38]. In another article, the injection of CGRP into trigeminal ganglia of female and male rats caused the development of periorbital mechanical and hindpaw allodynia, light sensitivity, and anxiety-like behaviour [90]. Additionally, the intra-trigeminal ganglionic CGRP administration causes orofacial heat hyperalgesia [91]. Moreover, the migraine-like mouse model induced by systemic GNT administration was able to increase H_2_O_2_ and 4-HNE levels in trigeminal ganglion and caused PMA via TRPA1 activation.

Administration of apocynin (a NADPH oxidase inhibitor) and α-lipoic acid (sequestrant of reactive oxygen species) reversed the GNT-evoked PMA [8]. These anti-oxidants might lead to reduced TRPA1 activation through the inhibition of oxidative stress generation, attenuating the migraine-like symptoms [8]. Moreover, the antinociceptive effect of α-lipoic acid had been described to attenuate the neuroinflammation in PMS patients by immunomodulatory and anti-inflammatory effects [92,93]. Previously, we also showed the antiallodynic effect of apocynin and α-lipoic acid (i.g. injection) in neuropathic pain-like behaviours induced by relapsing-remitting (RR) EAE and PMS-EAE, two different MS mouse models. The antinociceptive effects of these compounds was detected for 1–2 h in these MS models [43,44]. Apocynin and α-lipoic acid, when administered in mice in the doses used in this study, showed an antinociceptive effects for 1–2 h in other pain models [9,94]. Here, in a PMA model caused by PMS-EAE, we also observed the antiallodynic effect of apocynin and α-lipoic acid. Therefore, the blockage of TRPA1 or the reduction in TRPA1 endogenous agonists may be a valuable way to reduce PMA observed in MS.

## 4. Materials and Methods

### 4.1. Animals

Experiments and tissue collection were carried out according to the Animal Experiment Reports of Experiences In Vivo (ARRIVE) guidelines [95]. The animal care procedures were conducted under the University of Florence (protocol #1194/2015-PR) and the University of Santa Maria (protocols #8640200617/2017 and #6412121218/2018) ethical approval. C57BL/6 mice (female, 20–30 g, age 8–10 weeks), littermate wild-type (*Trpa1^+/+^*) and TRPA1-deficient (*Trpa1^−/−^*) mice (female, 25–30 g, age 8–10 weeks), generated by heterozygotes on a C57BL/6 background (B6; 29PTrpa1tm1Kykw/J; Jackson Laboratories) [96] were used. Female mice were used in this protocol following previous studies that optimised EAE model with female sex mice [97,98]. The animals were housed 5–10 per cage, with free access to standard animal feed (Puro Lab 22 PB granular form, Puro Trato, Brazil, and Charles River, Milan, Italy) and tap water. Room temperature (22 ± 1 °C) and humidity level (55%–65%) were controlled, and a 12-h light/dark cycle (lights from 7:00 am to 7:00 pm). The same researcher performed all measurements and was blinded to the administration of the medication or the group (control or PMS-EAE) to be tested.

### 4.2. Reagents

All experimental reagents, if not specified in the text, were purchased from Sigma-Aldrich (St. Louis, MO, USA). Mouse myelin oligodendrocyte glycoprotein (MOG_35–55_) was synthesized by EZBiolab (Carmel, CA, USA). Sumatriptan succinate (Sumax, 50 mg/tablet, Libbs Pharmaceutic LTDA, São Paulo, SP, Brazil) was obtained from the commercial pharmacy. *Mycobacterium tuberculosis* extract H37Ra was purchased from Difco Laboratories (Detroit, MI, USA).

### 4.3. PMS-Experimental Autoimmune Encephalomyelitis (PMS-EAE) Model Induction

To induce a mouse model of PMS-EAE, the mice were immunised with an emulsion of 200 μg of MOG_35-55_ (PT0213190101, EzBiolab) dissolved in phosphate-buffered saline (PBS) (100 µL) and Freund’s adjuvant (CFA) (F5506, Sigma-Aldrich, St. Louis, MO, USA) oil supplemented (100 µL) with 400 μg of *Mycobacterium tuberculosis* H37Ra extract (231141, Difco laboratories). The emulsion obtained for the reagent mixture was administered by a subcutaneous (s.c.) route in the flank region. Subsequently, all animals received a dose of 300 ng of pertussis toxin (P7208, Sigma-Aldrich, St. Louis, MO, USA) intraperitoneally (100 µL i.p.), which was re-administered 48 h after the induction. In the non-immunised animals (controls) MOG_35-55_ was not added in the emulsion mixture [98,99].

### 4.4. Assessment of Clinical Signs of PMS-EAE Model

The clinical signs of the PMS-EAE model were measured using a clinical scale that evaluated the neurological impairment using scores [100]. Then, animals were assessed using this scale: grade 0, normal mouse; grade 1, flaccid tail (disease onset); grade 2, mild hindlimb weakness with quick righting reflex; grade 3, severe hindlimb weakness with slow righting reflex; and grade 4, hindlimb paralysis in one hindlimb or both. Mice were monitored on different days p.i. of the PMS-EAE model (3–14 days) to assess the clinical signs and weight of PMS-EAE [44,100].

The animals were trained on the rotarod apparatus one day before induction. This rotarod training consisted of placing the animal on the spinning cylinder for 60 s at a fixed speed of 16 rpm, observing the latency to fall from the apparatus. This session was repeated two times. The rotarod test was performed on days 3, 5, 7, 9, 11, 13, and 14 p.i. using the same fixed speed for 180 s, and the animal’s latency to fall was recorded [97,100,101].

Animals showing a weight loss of 20% of the initial weight were excluded from the experiment. Additionally, mice displaying a clinical-grade ≥2, or that failed to remain 180 s in the rotarod, were removed from the study [101]. For the current study, all the induced animals developed a clinical score of 1 and nociceptive behaviours without significant weight loss until day 14 after induction. In this sense, it was unnecessary to exclude animals from the experimental set, as previously described [44].

### 4.5. Periorbital Mechanical Threshold Evaluation

First, the mice were placed individually in a restraint apparatus designed for evaluation of the periorbital mechanical threshold, for which the von Frey up-and-down paradigm was used [9]. Filaments of different stiffness were applied to the periorbital region of the animal, ranging from 0.008 to 0.6 g (g), starting with filament 0.07 g (0.008, 0.02, 0.04, 0.07, 0.16, 0.4, and 0.6 g) [9]. This paradigm continued for six measurements, or until four consecutive positive or four consecutive negative responses were seen. The periorbital mechanical withdrawal threshold response (in g) was then calculated from the resulting scores [102]. The animals were acclimatised for 60 min before the test, and all animals were assessed before PMS-EAE induction (baseline values). The periorbital mechanical threshold was evaluated on days 3, 5, 7, 9, 11, 13, and 14 p.i. in PMS-EAE or control animals. After 14 days (time 0) of PMS-EAE induction, the periorbital mechanical threshold was evaluated before treatments and from 1 up to 4 h after treatment by intragastric (i.g.) or intraperitoneal (i.p.) administration [44].

### 4.6. Treatment Protocols

Sumatriptan (Sumax, 50 mg/tablet, Libbs Pharmaceutic LTDA, São Paulo, SP, Brazil) is a drug frequently used in the clinical treatment of acute migraine [103,104]. This compound was administered to PMS-EAE induced and control mice by oral gavage (0.6 mg/kg, i.g.) to evaluate its antiallodynic effect [22,23]. Currently, gepants have emerged as the newest anti-migraines used in clinical treatment [54]. Thus, we chose to test olcegepant (SML2426, Sigma-Aldrich, St. Louis, MO, USA), the first gepant discovered, also named BIBN4096BS, administered by intraperitoneal injection (1 mg/kg, i.p.) or its vehicle (4% DMSO plus 4% Tween 80 (P1754, Sigma-Aldrich, St. Louis, MO, USA) in isotonic saline 0.9%) [8] in PMS-EAE and control mice, respectively.

The antinociceptive effect of selective TRPA1 antagonists, HC-030031 (H4415, Sigma-Aldrich, St. Louis, MO, USA) 2-(1,3-Dimethyl-2,6-dioxo-1,2,3,6-tetrahydro-7H-purin-7-yl)-N-(4-isopropylphenyl)acetamide (300 mg/kg i.g.) and A-967079 (SML0085, Sigma-Aldrich, St. Louis, MO, USA) (1E,3E)-1-(4-Fluorophenyl)-2-methyl-1-pentene-3-one oxime (100 mg/kg, i.g.), were also evaluated [9,83]. The animals were also treated with the non-selective TRPA1 antagonists metamizole (SML1488, Sigma-Aldrich, St. Louis, MO, USA; 100 mg/kg, i.g.) or propyphenazone (PHR 1389, Sigma-Aldrich, St. Louis, MO, USA; 100 mg/kg, i.g.) to evaluate the antinociceptive effect of these compounds in this model [58].

The animals were also treated with two anti-oxidants: α-lipoic acid (T5625, Sigma-Aldrich; sequestrant of reactive oxygen species, 100 mg/kg, i.g.) and apocynin (W508454, Sigma-Aldrich, St. Louis, MO, USA; NADPH oxidase inhibitor, 100 mg/kg, i.g.) by oral gavage [9,43,44]. All treatments, except olcegepant, had a vehicle group that received the correspondent vehicle (dimethyl sulfoxide-DMSO (472301, Sigma-Aldrich, St. Louis, MO, USA) 1% in isotonic saline 0.9%, 10 mL/kg) and periorbital mechanical threshold evaluated from 1 up to 3 or 4 h after vehicle or compound administration. All the treatments were administered in mice in a volume of 10 mL/kg, and the doses used were related to previous references as described.

### 4.7. Assessment of Oxidative Stress Parameters

The animals were euthanised after the behavioural testing on day 14 after PMS-EAE induction and trigeminal ganglia was dissected. The samples were homogenised in Tris-HCl buffer (10812846001, Sigma-Aldrich, St. Louis, MO, USA; 50 mM, pH 7.4) and centrifuged at 3000 rpm 4 °C for 10 min. The protein quantification was then performed by Bradford assay (Bovine Serum Albumin: A2153, Sigma-Aldrich, St. Louis, MO, USA; Comassie blue: 27815, Sigma-Aldrich, St. Louis, MO, USA) [105].

#### 4.7.1. TRPA1 Agonist (4-HNE and H_2_O_2_) Levels Measurement

The concentrations of H_2_O_2_ and 4-HNE (endogenous TRPA1 agonists) were analysed in the trigeminal ganglion after induction of PMS-EAE or its respective control. To determine the content of H_2_O_2_ the phenol red-horseradish peroxidase (HRPO) (P8250, Sigma-Aldrich, St. Louis, MO, USA) method was used [106]. Briefly, 25 mM of sodium azide (sodium azide: 1384, Dinâmica) was added to supernatants to inhibit the cytochrome c oxidase enzyme present in samples [107]. The homogenate containing sodium azide was centrifuged at 12,000 *g* for 20 min at 4 °C. A mixture containing supernatant, 25 μL of phenol red (114529, Sigma-Aldrich, St. Louis, MO, USA; 100 mg/mL), and 5 μL of HRPO (50 mg/ml) was incubated in the dark for 10 min at 25 °C. The reaction was stopped by NaOH (1 M; S5881, Sigma-Aldrich, St. Louis, MO, USA). The absorbance of the enzymatic reaction was read at 610 nm using a SpectraMax i3 ^®^ Platform (Molecular Devices, LLC, San Jose, CA, USA) microplate reader. H_2_O_2_ levels are expressed as H_2_O_2_ nanomoles (nmol H_2_O_2_) per mg protein compared to a standard H_2_O_2_ sample.

The levels of 4-HNE in the same samples were analysed using OxiSelect^TM^ H.N.E. adduct Competitive Elisa kit (STA-838, Cell Biolabs, INC, San Diego, CA USA); the samples were homogenised according to the kit specifications and measured by immunofluorescence [9]. The levels of 4-HNE were expressed as 4-HNE levels per mg protein when compared to the control group.

#### 4.7.2. Assessment of the Activity of NADPH Oxidase

An assay kit was used to analyse the activity of NADPH oxidase in the trigeminal ganglion samples (MAK038 CY0100, cytochrome c reductase, NADPH Sigma-Aldrich, Milan, Italy). The NADPH oxidase activity was expressed as U/ml/mg of protein.

### 4.8. Statistical Analyzes

Data were expressed as mean + standard error mean (SEM) and analysed statistically by Student’s t-test, one-way or two-way ANOVA according to the experimental protocol, followed by Bonferroni post-test when appropriate. The I_max_ was calculated using the following formula: 100 * (hour post-treatment–basal post-induction mean)/(basal pre-induction mean–basal post-induction mean). The individual values were inserted as column statistics in Prism GraphPad ^®^ and calculated the mean of these values. To meet parametric assumptions, data of mechanical threshold scores were log-transformed before analyses. Differences among groups were considered significant when *p* values were less than 0.05 (*p* < 0.05), using the GraphPad Prism 6.0 software.

## 5. Conclusions

Thus, this model of PMS-EAE caused PMA and this measure could be used to explore novel mechanisms for the control of facial nociception, such as that found in headache and migraine. The TRPA1 involvement in periorbital nociception was detected using TRPA1 antagonists administration and TRPA1 genetic deletion. Hence, this channel seems a valuable therapeutic target for PMS-EAE-induced nociception, such as neuropathic pain and PMA. Lastly, it was observed that increases in TRPA1 endogenous agonists and anti-oxidant treatments were effective in reducing the PMA in the PMS-EAE induced mice. These results suggest that the generation of TRPA1 endogenous agonists in the PMS-EAE mouse model may sensitise TRPA1 in trigeminal nociceptors to elicit PMA. It is important to clarify that here we are using a rodent model of PMS and it is necessary to conduct more investigation to relate the TRPA1 channels to future new treatments for headache in PMS patients.

## Figures and Tables

**Figure 1 pharmaceuticals-14-00831-f001:**
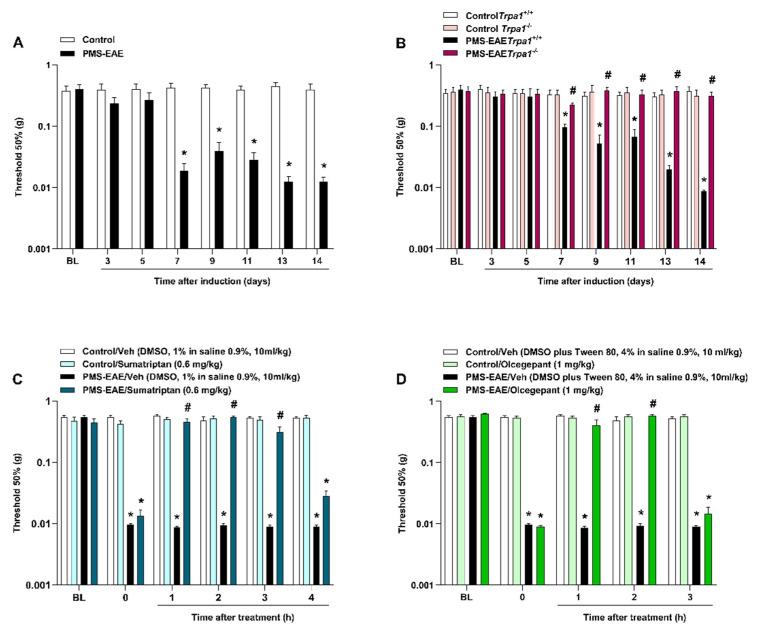
Progressive multiple sclerosis model by experimental autoimmune encephalomyelitis (PMS-EAE) caused PMA development, but *Trpa1^−/−^* mice induced to PMS-EAE did not develop PMA and antimigraine agents’ treatment reduced periorbital nociception. (**A**) Development of PMA 7 to 14 days after induction of PMS-EAE. (**B**) Absence of PMA in TRPA1 knockout mice after PMS-EAE induction when compared to control (*Trpa1^+/+^).* Intragastric (i.g.) administration of (**C**) sumatriptan (0.6 mg/kg) and intraperitoneal (i.p.) injection of (**D**) olcegepant (1 mg/kg) 14 days post-induction (p.i.) reduced the PMA generated by the PMS-EAE model. The vehicle group received dimethyl sulfoxide-DMSO 1% in isotonic saline 0.9% i.g. (sumatriptan) or 4% DMSO plus 4% Tween 80 in isotonic saline 0.9% i.p. (olcegepant). Basal (BL) values were observed before PMS-EAE induction. Data are expressed as mean + SEM (n = 6–8) in the graphs. * *p* < 0.05, when compared to the control group or baseline values (Two-way ANOVA, followed by Bonferroni’s post hoc test); and ^#^ *p* < 0.05 when compared with PMS-EAE vehicle-treated group (Two-way ANOVA, followed by Bonferroni’s post hoc test)].

**Figure 2 pharmaceuticals-14-00831-f002:**
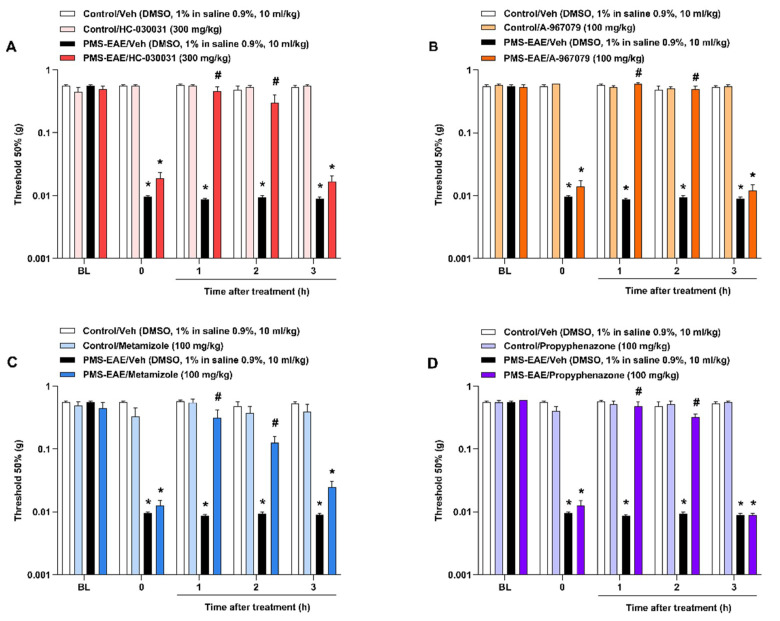
TRPA1 antagonists attenuated periorbital mechanical allodynia (PMA) in the progressive multiple sclerosis model induced by experimental autoimmune encephalomyelitis (PMS-EAE) in mice. All drugs were administered on day 14 post-induction (p.i.), and antinociceptive effects were observed from 1 up to 3 h after intragastric (i.g.) treatment. (**A**) HC-030031 (300 mg/kg), (**B**) A-967079 (100 mg/kg), (**C**) metamizole (100 mg/kg), and (**D**) propyphenazone (100 mg/kg) were tested for PMA induced by PMS-EAE. The vehicle group received dimethyl sulfoxide (DMSO) 1% in isotonic saline 0.9% by i.g. administration. Baseline values (BL) were observed before PMS-EAE induction. Data are expressed as mean + SEM (*n* = 6–8) in the graphs. * *p* < 0.05, when compared to the control group or baseline values; ^#^ *p* <0.05, when compared to the group treated with the PMS-EAE vehicle group (Two-way ANOVA, followed by the Bonferroni post hoc test).

**Figure 3 pharmaceuticals-14-00831-f003:**
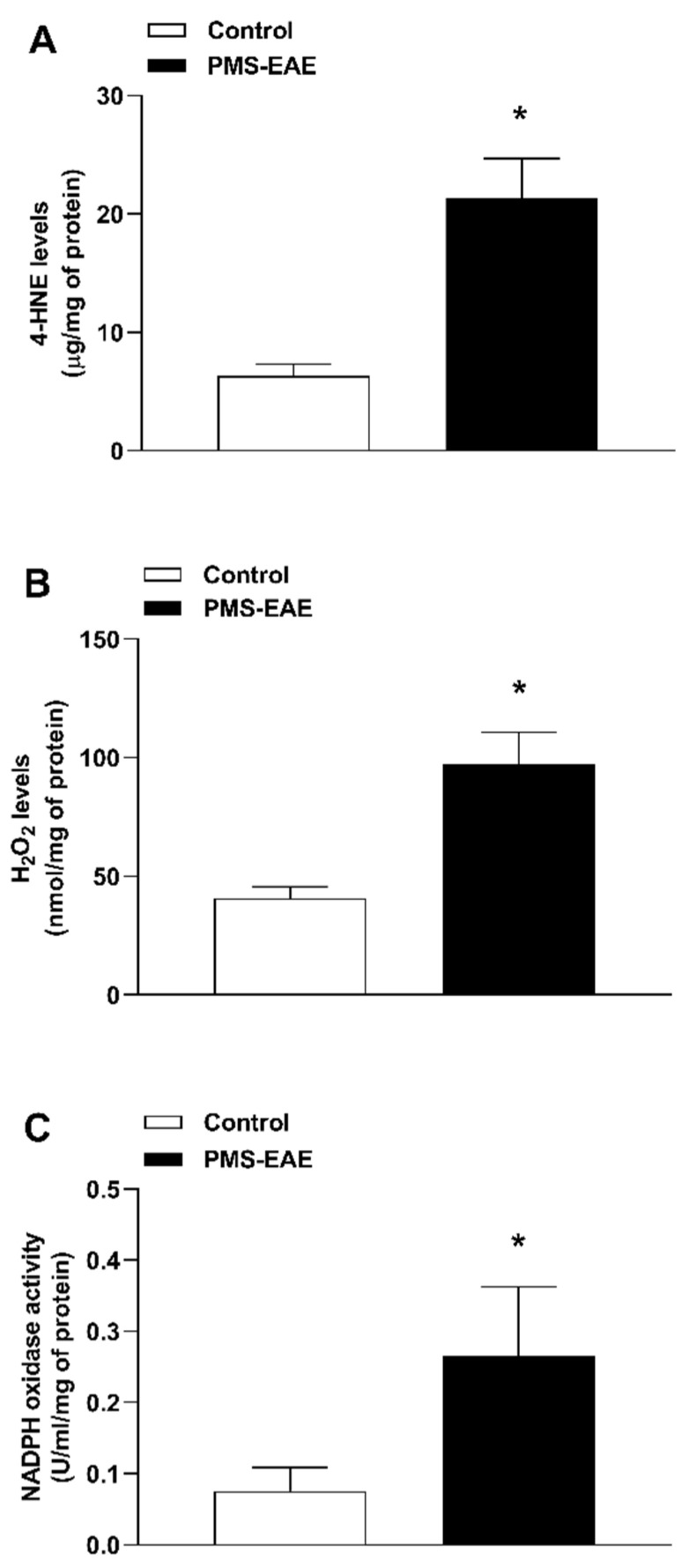
TRPA1 agonist levels and NADPH oxidase activity were increased in the trigeminal ganglion of the progressive multiple sclerosis model experimental autoimmune encephalomyelitis (PMS-EAE) induced-mice. Measurement of (**A**) 4-hydroxynonenal (4-HNE) and (**B**) H_2_O_2_ levels, (**C**) NADPH oxidase of trigeminal ganglion samples 14 days after PMS-EAE mice induction. Data are expressed as mean + SEM (*n* = 6–7). * *p* < 0.05, when compared to the control group parametric Student’s *t*-test.

**Figure 4 pharmaceuticals-14-00831-f004:**
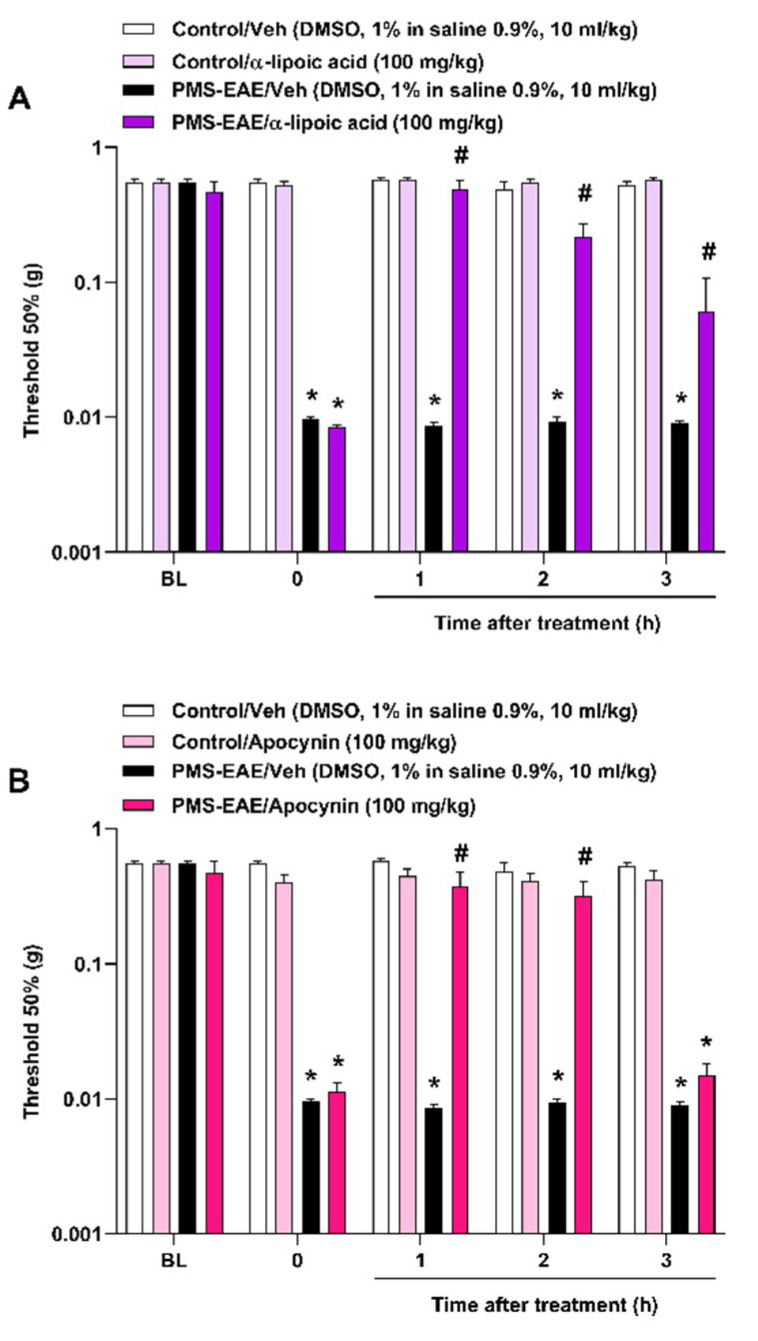
The intragastric treatment with α-lipoic acid and apocynin attenuated periorbital mechanical allodynia (PMA) in the progressive multiple sclerosis model induced by experimental autoimmune encephalomyelitis (PMS-EAE) in mice. The administration of (**A**) α-lipoic acid (100 mg/kg) or (**B**) apocynin (100 mg/kg) intragastrically (i.g.) was performed on day 14 post-induction (p.i), and their antinociceptive effects were observed from 1 up to 3 h. The vehicle group received dimethyl sulfoxide (DMSO) 1% in isotonic saline 0.9% i.g. Baseline values (BL) were observed before PMS-EAE induction. Data are expressed as mean + SEM (*n* = 6–8) in the graphs. * *p* < 0.05 when compared to the control group or baseline values; ^#^ *p* < 0.05 when compared to the PMS-EAE group treated with the vehicle (Veh) (Two-way ANOVA followed by Bonferroni’s post hoc test).

## Data Availability

Data is contained within the article and Appendix A.

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
