# Peer review of "Periorbital Nociception in a Progressive Multiple Sclerosis Mouse Model Is Dependent on TRPA1 Channel Activation"

_pharmaceuticals, 2021, doi:10.3390/ph14080831_

Round 1

Reviewer 1 Report

In the manuscript entitled “Periorbital nociception in a progressive multiple sclerosis mouse model is dependent on TRPA1 channel activation”, Dalenogare and co-workers examined the function of TRPA1 channels in a murine model of progressive multiple sclerosis. Here they show that TRPA1-/- mice fail to induce the periorbital mechanical allodynia and the intragastric administration of TRPA1 receptor antagonists reverse the PMS-EAE-induced pain behavior. They further provide the evidence that PMS-EAE increased endogenous TRPA1 channel agonist in trigeminal ganglion and antioxidants attenuated mechanical allodynia in PMS-EAE model. Several concerns as indicated below need to be addressed to improve the quality of the manuscript.

Major concerns

In general, EAE produces a neuropathic pain and which is explained by authors in the introduction section and their previous paper published in Mol Neurobiology 2020. However, it has been shown that EAE produced trigeminal neuropathic pain and facial hypersensitivity (Thorburn K, Pain 2016). In here, the authors use a PMS-EAE model as a headache-like model through the manuscript. Even though, the authors confirm the reduction of mechanical threshold in PMS-EAE, it is over-statement to claim that PMS-EAE is a headache-like model. I think a main finding in the current paper has been addressed in recent their publications (Dalenogare, Experimental Neurology, 2020). At least, the authors need to differentiate whether pain is due to the orofacial neuropathic pain or headache-like pain if their conclusion is a treatment of headache.

Second, the authors need to provide every experimental data which is written in the section of materials and methods. The results of “4.4 Assessment of clinical signs of PMS-EAE model” are missing in the manuscript and it is important to show the evaluation of EAE model they used.

Specific

1) What is the reason the authors measured TRPA1 agonists (4HNE and H2O2) in the trigeminal ganglion? TRPA1 channels act on the peripheral and central nerve terminals to detect or modulate the pain. They used a von Frey test which stimulates skin, and antioxidants treatments attenuate a mechanical allodynia. How increased reactive oxygen species in trigeminal ganglia (soma) contributes to generate a pain which evoked in peripherally?

2) Please be consistent with the terms ”TRPA+/+” or “TRPA1+/+”.

3) Please explain what is alpha-lipoic acid and apocynin in the manuscript. The authors mention “antioxidants” in the Abstract only.

Author Response

Manuscript Pharmaceuticals - 1314307

ANSWERS TO THE COMMENTS OF REVIEWER ‘1’

Question 1: “In general, EAE produces a neuropathic pain, and which is explained by authors in the introduction section and their previous paper published in Mol Neurobiology 2020. However, it has been shown that EAE produced trigeminal neuropathic pain and facial hypersensitivity (Thorburn K, Pain 2016). In here, the authors use a PMS-EAE model as a headache-like model through the manuscript. Even though, the authors confirm the reduction of mechanical threshold in PMS-EAE, it is over-statement to claim that PMS-EAE is a headache-like model. I think a main finding in the current paper has been addressed in recent their publications (Dalenogare, Experimental Neurology, 2020). At least, the authors need to differentiate whether pain is due to the orofacial neuropathic pain or headache-like pain if their conclusion is a treatment of headache.”

Answer: Recently, it was described the development of trigeminal neuropathic pain behaviors and facial hypersensitivity in a PMS-EAE model in mice using the air puff testing (doi: 10.1097/j.pain.0000000000000409), or cotton glove stimulation and von Frey test (doi: 10.3389/fimmu.2016.00369) in the whisker pad area. When the whisker pad area is stimulated, some lesion or inflammation process in the trigeminal nerve is being analyzed (doi: 10.1016/j.neuroscience.2015.03.051; doi: 10.1016/j.jneumeth 2011.07.006), as it was previously cited in the references of these two articles. However, in our study, we performed the von Frey test in the periorbital area, which was demonstrated as a headache and migraine-related area for the development of mechanical allodynia, as described in recent studies (doi: 10.1186/s10194-019-0968-1; doi: 10.1093/brain/awy177; doi: 10.1523/JNEUROSCI.0364-19.2019; doi: 10.1177/0333102418779557).

Moreover, trigeminal neuropathic pain committed a minor part of multiple sclerosis patients (from 1 to 10%) (doi: 10.1038/nrneurol.2011.120; doi: 10.1007/s11916-019-0800-2). After all, it was extensively described the presence of two main types of secondary headaches in multiple sclerosis patients (migraine and tension-type headache) (doi: 10.1007/s10072-009-0053-7), which have a reported range from 4 to 61.8% (doi: 10.1007/s10072-009-0053-7).

In addition, the primary purpose of our first two articles was to investigate the relation between neuropathic pain behaviors induced by two different multiple sclerosis mouse models and activation of the TRPA1 channel in these models (doi: 10.1007/s12035-020-01891-9; doi: 10.1016/j.expneurol.2020.113241). Here, our main purpose was to optimize the development of periorbital mechanical allodynia in a progressive EAE model and to detect the role of TRPA1 in this nociceptive behavior. Unfortunately, we cannot distinguish if the periorbital mechanical allodynia observed in our results is caused by migraine or tension-type headache, so we revised all the terms used in the manuscript and changed it to “periorbital mechanical allodynia” instead of “headache-like model”. Thus, we added this information to the sections “Introduction” (Page 2, lines 47-53; lines 68-80) and “Discussion” (Page 8, lines 226-231) of our revised manuscript.

Question 2: “Second, the authors need to provide every experimental data which is written in the section of materials and methods. The results of “4.4 Assessment of clinical signs of PMS-EAE model” are missing in the manuscript and it is important to show the evaluation of the EAE model they used.”

Answer: As requested, we added supplementary data for the clinical disease score, rotarod test, and weight from control and PMS-EAE induced mice in the section “Supplementary Materials:” as “Figure S1”. In addition, we added a paragraph in the section “Material and Methods” about these results (Page 15, line 476-484).

Question 3: “What is the reason the authors measured TRPA1 agonists (4HNE and H2O2) in the trigeminal ganglion? TRPA1 channels act on the peripheral and central nerve terminals to detect or modulate the pain. They used a von Frey test which stimulates skin, and antioxidants treatments attenuate a mechanical allodynia. How increased reactive oxygen species in trigeminal ganglia (soma) contributes to generate a pain which evoked in peripherally?”

Answer: We choose to detect the levels of TRPA1 agonists in the trigeminal ganglion because different mouse models demonstrated a relation between facial allodynia and inflammatory/oxidative stress process in the trigeminal ganglia (doi: 10.1097/j.pain.0000000000000409; doi: 10.1177/0333102419896539; doi: 10.3390/ijms20030711; doi: 10.1093/brain/awy177).

Moreover, in a previous study using the same PMS-EAE mice model, it was observed the development of facial hypersensitivity when air puff was applied to the whisker pad area (doi: 10.1097/j.pain.0000000000000409). Also in this article, it was shown the T-cell infiltration and glial activation in primary trigeminal afferents (doi: 10.1097/j.pain.0000000000000409).

Besides, in another article, the injection of calcitonin-gene related peptide (CGRP) in trigeminal ganglia of rats causes the development of periorbital mechanical and hind paw allodynia, light sensitivity, and anxiety like-behavior (doi: 10.1177/0333102419896539). Also, the intra-trigeminal ganglionic CGRP administration causes orofacial heat hyperalgesia (doi: 10.3390/ijms20030711).

Moreover, the migraine-like mice model induced by systemic glyceryl nitrate (GNT) administration was able to increase nitric oxide, hydrogen peroxide, and 4-HNE levels in the trigeminal ganglion and caused periorbital mechanical allodynia via TRPA1 activation. The administration of apocynin (an NADPH oxidase inhibitor) and α-lipoic acid reversed the GNT-evoked allodynia (doi: 10.1093/brain/awy177).

Thus, observing all these findings, we chose to investigate whether the enhancement of TRPA1 agonists (hydrogen peroxide and 4-HNE) in the trigeminal ganglion of PMS-EAE induced mice could be one of the mechanisms to generate the periorbital mechanical allodynia. We included in the section “Discussion” of our revised manuscript this information (Page 10, lines 297-310).

Question 4: “Please be consistent with the terms “TRPA+/+” or “TRPA1+/+.”

Answer: As requested, we changed the term “TRPA+/+” to the correct term “TRPA1+/+”, all the alterations were highlighted in red in section “Results 2.1” (Page 3, line 102; Page 4, line 114).

Question 5: “Please explain what alpha-lipoic acid and apocynin in the manuscript. The authors mention “antioxidants” in the Abstract only.”

Answer: We highlighted in red the paragraph explaining the function of α-lipoic acid and apocynin in section “Methods 4.6” (Page 13, lines 414-416). As requested, we also added some information in section “Discussion” (Page 10, lines 311-325) about the effect of these compounds in nociceptive models. Besides, we included in the abstract some information about the mechanism of action of these compounds (Page 1, lines 34-35).

We hope that the new version of our manuscript has been satisfactorily improved and is now suitable for publication in Pharmaceuticals.

Sincerely yours,

Professor Gabriela Trevisan

Reviewer 2 Report

Overall Summary

The authors used behavioral pharmacology to test whether TRPA1 antagonists, established migraine therapeutics, or antioxidants alleviate periorbital mechanical allodynia (PMA) in a mouse model of progressive multiple sclerosis (experimental autoimmune encephalomyelitis induced by MOG35-55; PMS-EAE). They found that genetic knockout of TRPA1 prevents the development of PMA and the common TRPA1 antagonists HC-030031 and A-967079, as well as the partial TRPA1 antagonists metamizole and propyphenazone, reverse PMA 14d after PMS-EAE induction. The common anti-migraine therapeutics sumatriptan and olcegepant and the antioxidant compounds alpha-lipoic acid and apocynin also reversed PMA 14d after PMS-EAE induction. The data is nicely presented in the figures, suggesting an appropriate experimental approach. The lack of discussion of the short-lived effect of all drugs tested in this study and the outcome of evoked mechanical allodynia, as opposed to rodent behavioral assays that may better reflect the non-evoked debilitating spontaneous pain in human patients, temper excitement for publication. In addition, a poorly worded Discussion and numerous (but mostly minor) errors in the wording and content need to be addressed before this paper is acceptable for publication.

Major Comments

  • The anti-allodynic effect of sumatriptan and olcegepant was relatively short (gone within 4h). Same for other drugs used in this study. How would this translate to effective use in humans with MS-related headache? How does the anti-allodynic effect compare to the pharmacokinetics/pharmacodynamics for these drugs? Please include this type of discussion in the revision.
  • What is the rationale for doses and routes of administration? Are these doses and routes of administration related to that in human patients? Including this information is necessary to support the conclusion that TRPA1 is a potential therapeutic target for treatment of headache in MS patients.
  • Numerous grammar and typographical errors in the writing. Discussion is very difficult to read. English language editing service is suggested. Quality of writing is below that of recent previous publications from this group.
  • Since mechanical thresholds are not shown beyond 14d of PMS-EAE induction, it is inappropriate to discuss the peak of nociceptive behaviors. In some RR-EAE models, the pain and clinical scores may recover and then worsen. Is this true for the mode of EAE induction used in the current study? If so, please show the data.
  • Clinical scoring and Rotarod are described in the Methods section but no Results/data are shown. If this is a model of progressive MS using EAE, then clinical signs should worsen over time (correct?), but no data is provided. Please include this data.
  • This is not the first study to show antiallodynic effects of TRPA1 antagonists in a PMS-EAE model. Please revise discussion accordingly to include these other studies and/or specify that you are (probably) referring to PMA (facial pain-like behavior) as opposed to allodynia of the hands or feet.
  • Is periorbital mechanical allodynia an established outcome of headache-like pain in rodents? If so, please include this background information. It is unclear how mechanical allodynia reflects MS-related headache pain since typical pain in patients is non-evoked. It would be worth including assays of non-evoked pain in this PMS-EAE model to further support TRPA1 as a potential therapeutic target in PMS patients (e.g., conditioned place preference).
  • The Introduction states that headache prevalence in patients with multiple sclerosis is greater in women and this study uses only female mice. This should be expanded upon in the Discussion. For example, are treatments for MS-related headache different in men versus women? Otherwise, including this sex-difference information in the Introduction is distracting and misleading (suggesting that this is a study comparing sex as a biological variable). It is also unclear whether male or female TRPA1+/+ and TRPA1-/- were used, please clarify.
  • There are other endogenous agonists of TRPA1 that may be upregulated in MS (e.g. methylglyoxal, acrolein). It may be useful to discuss the possibility that other compounds besides 4-HNE and H2O2 may be involved in PMA induced by PMS-EAE. There are also papers showing the involvement of TRPA1 in demyelination and neurobehavioral deficits, it may be useful to include discussion of these related studies.

Minor Comments

  • Figure 1B make TRPA+/+ bold for consistency.
  • Include product numbers for all drugs/materials used. Use RRID if applicable.
  • Modify Methods 4.3 to clarify the volume and timing of s.c. and i.p. injections.
  • Why would animals showing clinical score greater than 2 and impaired motor coordination (rotarod disturbance) be excluded from the study? This doesn’t make sense if the aim is to study periorbital pain in a model of progressive MS.
  • Methods 4.7.1 is it twelve thousand times g or twelve times g? Use a comma instead of period?
  • 4-HNE levels were standardized to mg protein. Please include detail of protein quantification in trigeminal ganglia.
  • Please double-check Imax values for metamizole and propyphenazone in Figure 2.
  • Define “RR-EAE”.
  • Units for 4-HNE levels in Figure 3A are incomplete.
  • Conclusions are just restatement of Results/Discussion. Please revise.
  • Methods states there was no Vehicle control group for olcegepant but there is clearly a Vehicle group present in Figure 1D. In addition, it appears the Vehicle given to control mice was different than that given to PMS-EAE mice. Please clarify.

Author Response

Manuscript Pharmaceuticals – 1314307

ANSWERS TO THE COMMENTS OF REVIEWER ‘2’

“The authors used behavioral pharmacology to test whether TRPA1 antagonists, established migraine therapeutics, or antioxidants alleviate periorbital mechanical allodynia (PMA) in a mouse model of progressive multiple sclerosis (experimental autoimmune encephalomyelitis induced by MOG35-55; PMS-EAE). They found that genetic knockout of TRPA1 prevents the development of PMA and the common TRPA1 antagonists HC-030031 and A-967079, as well as the partial TRPA1 antagonists metamizole and propyphenazone, reverse PMA 14d after PMS-EAE induction. The common anti-migraine therapeutics sumatriptan and olcegepant and the antioxidant compounds alpha-lipoic acid and apocynin also reversed PMA 14d after PMS-EAE induction. The data is nicely presented in the figures, suggesting an appropriate experimental approach. The lack of discussion of the short-lived effect of all drugs tested in this study and the outcome of evoked mechanical allodynia, as opposed to rodent behavioral assays that may better reflect the non-evoked debilitating spontaneous pain in human patients, temper excitement for publication. In addition, a poorly worded Discussion and numerous (but mostly minor) errors in the wording and content need to be addressed before this paper is acceptable for publication.”

Question 1: “The anti-allodynic effect of sumatriptan and olcegepant was relatively short (gone within 4h). Same for other drugs used in this study. How would this translate to effective use in humans with MS-related headache? How does the anti-allodynic effect compare to the pharmacokinetics/pharmacodynamics for these drugs? Please include this type of discussion in the revision.”

Answer: As requested, we added in section “Discussion” pharmacokinetics/pharmacodynamics data for these drugs about the compounds studied sumatriptan (Page 8, lines 208-213); olcegepant (Page 8, lines 219-225); metamizole and propyphenazone, HC-030031 and A967079 (Page 9, lines 236-250); Apocynin and alpha-lipoic acid (Page 10-11, lines 311-325). In addition, we highlighted all the treatment references in the section “Methods 4.6”, which were previously demonstrated their effects in different pain models in mice and present a similar curve of time versus effect results (doi: 10.1002/ijc.31911; doi: 10.1186/s12915-020-00935-9; doi: 10.1016/j.bbi.2020.04.037; doi: 10.1093/brain/aww038; doi: 10.1007/s12035-020-01891-9; doi: 10.1016/j.expneurol.2020.113241) (Page 13, lines 401-421).

Question 2: “What is the rationale for doses and routes of administration? Are these doses and routes of administration related to that in human patients? Including this information is necessary to support the conclusion that TRPA1 is a potential therapeutic target for treatment of headache in MS patients.”

Answer: Firstly, only metamizole (dipyrone), propyfenazone, and sumatriptan are currently used to treat pain in humans, all these compounds have oral routes (doi: 10.1002/14651858.CD004842.pub3.; doi: 10.1191/0960327104ht439oa; doi: 10.1111/head.12499). The olcegepant was the first gepant developed but it was discontinued because of difficulties in developing an oral formulation (doi: 10.1080/13543784.2019.1618830) and is currently used as a CGRP antagonist for pathophysiology investigations in migraine-like rodent models (doi: 10.1093/brain/awy177; doi: 10.1093/brain/awr272) in section “Discussion” (Page 8, lines 219-223). Other compounds are pharmacological approaches to test the role of TRPA1 and oxidative stress in headache-like pain in PMS-EAE induction. At least selective TRPA1 antagonist was not approved for clinical use in humans (doi: 10.1080/13543776.2020.1797679). As request, we highlighted all the references for doses and routes for the compounds used in this experiments and they are added in section “Methods 4.6” (Page 13, lines 401-421): sumatriptan (doi: 10.1177/0333102415584313; doi: 10.1177/0333102418779548), olcegepant (doi: 10.1093/brain/awy177), HC-030031 (doi: 10.1002/ijc.31911; doi: 10.1002/ijc.31911); A-967079 (doi: 10.1002/ijc.31911; doi: 10.1002/ijc.31911); metamizole (doi: 10.1111/bph.13129); propyphenazone (doi: 10.1111/bph.13129); alpha-lipoic acid (doi: 10.1002/ijc.31911; doi: 10.1007/s12035-020-01891-9; doi: 10.1016/j.expneurol.2020.113241), and apocynin (doi: 10.1002/ijc.31911; doi: 10.1007/s12035-020-01891-9; doi: 10.1016/j.expneurol.2020.113241).

Question 3: “Numerous grammar and typographical errors in the writing. Discussion is very difficult to read. English language editing service is suggested. Quality of writing is below that of recent previous publications from this group.”

Answer: Unfortunately, due to a short deadline to answer the reviewers, we could not send our manuscript to an English language editing service, as it was suggested. However, we revised the manuscripts using Grammarly, and the discussion section was rewritten. Besides, if necessary, we could send the manuscript to an English language editing service.

Question 4: Since mechanical thresholds are not shown beyond 14d of PMS-EAE induction, it is inappropriate to discuss the peak of nociceptive behaviors. In some RR-EAE models, the pain and clinical scores may recover and then worsen. Is this true for the model of EAE induction used in the current study? If so, please show the data.

Answer: Previous data demonstrated the peak of hind paw mechanical and cold allodynia around 14 days p.i. The clinical score peak was reached at 19 days p.i. and was diminished at 21 days p.i. (doi: 10.1016/j.pain.2008.11.002). Moreover, we also demonstrated the evaluation of clinical scores in our first publication using this same model, which showed an increase at 17 p.i. (doi: 10.1007/s12035-020-01891-9). Finally, we agree to exclude the term “peak of nociception behaviors”, because we did not measure the periorbital mechanical allodynia beyond 14 days post-induction. However, as the animals after 14 days have a motor impairment and weight loss, we have selected this time point to perform our study, as our previous publication using this PMS-EAE model (doi: 10.1007/s12035-020-01891-9). Besides, the periorbital mechanical threshold of PMS-EAE mice was 0.0126 ± 0.0002 g and for control was 0.4273 ± 0.2177 g 14 days p.i., which shows that PMS-EAE induced mice had reduced mechanical threshold considering that the lowest von Frey filament used was 0.008 g. We added this information in the sections “Results” (Page 3, lines 90-100) and “Discussion” (Page 8, lines 196-198) section.

Question 5: “Clinical scoring and Rotarod are described in the Methods section, but no Results/data are shown. If this is a model of progressive MS using EAE, then clinical signs should worsen over time (correct?), but no data is provided. Please include this data.”

Answer: As requested, we added supplementary data for the clinical disease score, rotarod test, and weight from control and PMS-EAE mice in the section “Supplementary Materials:” as “Figure S1”. In addition, we added a paragraph in the section “Results 2.1” about these results and our previous publication reference (Page 3, lines 90-100)

Question 6: This is not the first study to show antiallodynic effects of TRPA1 antagonists in a PMS-EAE model. Please revise discussion accordingly to include these other studies and/or specify that you are (probably) referring to PMA (facial pain-like behavior) as opposed to allodynia of the hands or feet.

Answer: We performed the von Frey test in the periorbital area in this study protocol differently from our previous data (doi: 10.1007/s12035-020-01891-9; doi: 10.1016/j.expneurol.2020.113241), where this mechanical allodynia test was performed in the right hind paw. Thus, we maintain using the term “periorbital mechanical allodynia (PMA)” according to extensively use in the research about headache and facial nociception in rodent models (doi: 10.1186/s10194-019-0968-1; doi: 0.1186/s12974-019-1459-7; doi: 10.1177/0333102419896539; doi: 10.1093/brain/awy177). Thus, we revised the manuscript to clarify that this is the first study to detect the role of TRPA1 in PMA in this model of PMS-EAE.

Question 7: “Is periorbital mechanical allodynia an established outcome of headache-like pain in rodents? If so, please include this background information. It is unclear how mechanical allodynia reflects MS-related headache pain since typical pain in patients is non-evoked. It would be worth including assays of non-evoked pain in this PMS-EAE model to further support TRPA1 as a potential therapeutic target in PMS patients (e.g., conditioned place preference).”

Answer: In our study, we performed the von Frey test in the periorbital area, which was demonstrated as a headache and migraine-related area for the development of mechanical allodynia, as described in recent studies (doi: 10.1186/s10194-019-0968-1; doi: 10.1093/brain/awy177; 10.1523/JNEUROSCI.0364-19.2019; doi: 10.1177/0333102418779557; doi: doi.org/10.1016/j.pain.2012.06.012). After all, it was extensively described the presence of two main types of secondary headaches in multiple sclerosis patients (migraine and tension-type headache) (doi: 10.1007/s10072-009-0053-7), which have a reported range from 4 to 61.8% (doi: 10.1007/s10072-009-0053-7). Here, our primary purpose was to optimize the development of periorbital mechanical allodynia in a progressive EAE model and to detect the role of TRPA1 in this nociceptive behavior. Unfortunately, we cannot distinguish if the periorbital mechanical allodynia observed in our results is caused by migraine or tension-type headache, so we revised all the terms used in the sections and changed it to “periorbital mechanical allodynia” instead of a headache-like model. Thus, we added this information in the sections “Introduction” (Page 2, lines 47-53; lines 68-80) and “Discussion” (Page 8, lines 226-231) of our manuscript.

We did not demonstrate non-evoked nociceptive tests in this present study; but it is an essential suggestion for future investigations. However, due to the pandemic, it would not be possible to carry out further experiments as we will need to repeat all the administration protocols. We have inserted this information in the section “Discussion” about the mouse grimace scale and conditioned place preference usage to measure spontaneous pain in facial pain models (Page 8, lines 198-203).

Question 8: “The Introduction states that headache prevalence in patients with multiple sclerosis is greater in women and this study uses only female mice. This should be expanded upon in the Discussion. For example, are treatments for MS-related headache different in men versus women? Otherwise, including this sex-difference information in the Introduction is distracting and misleading (suggesting that this is a study comparing sex as a biological variable). It is also unclear whether male or female TRPA1+/+ and TRPA1-/- were used, please clarify.”

Answer: As requested, we added information about headache treatment in the section “Discussion” (Page 7, lines 179-187)”. We also deleted the sex-difference information from the introduction section.

To clarify, we add the information about the sex of TRPA1 ko mice, and it was described that all the animals used in this experiment were female because of the EAE optimization reference model (doi: 10.1007/s12035-016-0014-0; doi: 10.1016/j.pain.2010.03.037). The additional information is in the section “Method 4.1” (Page 11, lines 333-337).

Question 9: “There are other endogenous agonists of TRPA1 that may be upregulated in MS (e.g. methylglyoxal, acrolein). It may be useful to discuss the possibility that other compounds besides 4-HNE and H2O2 may be involved in PMA induced by PMS-EAE. There are also papers showing the involvement of TRPA1 in demyelination and neurobehavioral deficits; it may be useful to include discussion of these related studies.”

Answer: As recommended, about the involvement of TRPA1 in demyelination and neurobehavioral deficits, we added one paragraph in section “Discussion” (Page 9, line 251-254). Besides, as requested, we added in section “Discussion” (Page 10, line 227-287) information about the role of other endogenous agonists of TRPA1 in multiple sclerosis.

Minor Comments

Question 10: Figure 1B make TRPA+/+ bold for consistency.

Answer: As requested, we corrected it in Figure 1B.

Question 11: Include product numbers for all drugs/materials used. Use RRID if applicable.

Answer: As recommended, we added this information about all compounds used in the revised version of our manuscript.

Question 12: Modify Methods 4.3 to clarify the volume and timing of s.c. and i.p. injections.

Answer: As requested, we added the information about the volume and timing administration, the MOG and CFA emulsion was administered only one time, and Pertussis toxin was administered two times, one on the induction day and one after 48 hours post-induction (Page 11-12, lines 354-361).

Question 13: Why would animals showing clinical score greater than 2 and impaired motor coordination (rotarod disturbance) be excluded from the study? This doesn’t make sense if the aim is to study periorbital pain in a model of progressive MS.

Answer: First, we followed this exclusion criteria because of the animal’s health care and according to ARRIVE protocols. Previous data demonstrated the peak of hind paw mechanical and cold allodynia around 14 days p.i. The clinical score peak was reached at 19 days p.i. and was diminished at 21 days p.i. (doi: 10.1016/j.pain.2008.11.002). Moreover, we also demonstrated the evaluation of clinical scores in our first publication using this same model, which showed an increase at 17 p.i. (doi: 10.1007/s12035-020-01891-9). Finally, we agree to exclude the term “peak of nociception behaviors”, because we did not measure the periorbital mechanical allodynia beyond 14 days post-induction. However, as the animals after 14 days have a motor impairment and weight loss, we have selected this time point to perform our study, as our previous publication using this PMA-EAE model (doi: 10.1007/s12035-020-01891-9; doi: 10.1016/j.pain.2008.11). Besides, the periorbital mechanical threshold of PMS-EAE mice was 0.0126 ± 0.0002 g and for control was 0.4273 ± 0.2177 g 14 days p.i., which shows that PMS-EAE induced mice had a reduced threshold considering that the lowest von Frey filament used was 0.008 g. We added this information in the sections “Results” (Page 3, lines 90-100) and “Discussion” (Page 8, lines 196-198).

Question 14: Methods 4.7.1 is it twelve thousand times g or twelve times g? Use a comma instead of period?

Answer: The expression of the centrifuge velocity is 12000 x (times) g that was highlighted in red in the section “Methods” (Page 13, line 435).

Question 15: “4-HNE levels were standardized to mg protein. Please include detail of protein quantification in trigeminal ganglia.”

Answer:  As requested, we added the following paragraph, Section “Materials and Methods” (Page 13, line 426-427): “After, the protein quantification was performed by Bradford assay (doi: 10.1006/abio.1976.9999)”.

Question 16: Please double-check Imax values for metamizole and propyphenazone in Figure 2.

Answer: We revised the Imax value and the calculation for these two compounds. Probably, the bars appear more than 69% because the graph is in the Log10 scale. We added this information about the scale in the section “Material and Methods” (Page 14, line 457-459).

Question 17: Define “RR-EAE”.

Answer: We added the definition of “RR-EAE”, highlighted in red section “Introduction” (Page 3, line 83), that was a relapsing-remitting multiple sclerosis model in mice induced by experimental autoimmune encephalomyelitis using the saponin Quil A as adjuvant.

Question 18: Units for 4-HNE levels in Figure 3A are incomplete.

Answer: As recommended, we corrected Figure 3A.

Question 19: “Conclusions are just restatement of Results/Discussion. Please revise.”

Answer: As requested, we added a paragraph in the section “Conclusion” (Page 14, line 463-473).

.

Question 20: “Methods states there was no Vehicle control group for olcegepant but there is clearly a Vehicle group present in Figure 1D. In addition, it appears the Vehicle given to control mice was different from that given to PMS-EAE mice. Please clarify.”

Answer: As requested, we corrected Figure 1D and added the description of the olcegepant vehicle in the section “Methods” (Page 13, line 404-407).

We hope that the new version of our manuscript has been satisfactorily improved and is now suitable for publication in Pharmaceuticals.

Sincerely yours,

Professor Gabriela Trevisan

Reviewer 3 Report

This study show the activation of TRPA1 channel and its signal pathway in a mouse model of periorbital nociception in a progressive multiple sclerosis. The manuscript is well written with detailed descriptions of methodology and results. The balanced discussion provides a good review of related work of others. The Figures are quite detailed. Both the topic of the study as well as the findings are interesting. However, I have some comments about manuscript, as described below:

  1. Please, explain why you used only one dose for the tested drugs (sumatriptan, olcegepant, metamizole and propifenazone) in the pain test? On what basis did you choose those doses? Why you used four different TRPA1 antagonists, both selective and nonselective antagonist?
  2. I think that it would be interesting to explain in more detail in the Discussion section the relationship between TRPA1 receptors and tested oxidative parameters in the used experimental model. Also, many factors can affect the activation of TRPA1 receptors, such as changes in tissue acidity, different ions… (Srebro et al., Neuroscience. 2016). This consideration should be mentioned in the Discussion section.

Author Response

Manuscript Pharmaceuticals - 1314307

ANSWERS TO THE COMMENTS OF REVIEWER ‘3’

“This study show the activation of TRPA1 channel and its signal pathway in a mouse model of periorbital nociception in a progressive multiple sclerosis. The manuscript is well written with detailed descriptions of methodology and results. The balanced discussion provides a good review of related work of others. The Figures are quite detailed. Both the topic of the study as well as the findings are interesting. However, I have some comments about manuscript, as described below:”

Question 1: Please, explain why you used only one dose for the tested drugs (sumatriptan, olcegepant, metamizole and propifenazone) in the pain test? On what basis did you choose those doses? Why you used four different TRPA1 antagonists, both selective and nonselective antagonist?

Answer: First, we usually optimized to do only one curve time versus response for nociception pain tests, as we showed in other published articles for our group: (doi: 10.1002/ijc.31911; doi: 10.1186/s12915-020-00935-9; doi: 10.1016/j.bbi.2020.04.037; doi: 10.1093/brain/aww038; doi: 10.1007/s12035-020-01891-9; doi: 10.1016/j.expneurol.2020.113241). We choose the treatment doses according to the following references, which were highlighted in red in the section “Methods 4.6” (Page 13, lines 400-421): Sumatriptan, (doi: 10.1177/0333102415584313; doi: 10.1177/0333102418779548); olcegepant, (doi :10.1093/brain/awy177), metamizole and propyphenazone (doi: 0.1111/bph.13129).

Lastly, we also tested two selective TRPA1 antagonists, HC-030031 and A-967079 in the other articles using the multiple sclerosis models (doi: 10.1007/s12035-020-01891-9; doi: 10.1016/j.expneurol.2020.113241). We decided to test metamizole and propyphenazone because of their use as analgesic in different form of pain in humans (doi: 10.1002/14651858.CD011421.pub2; doi: 10.1097/00045391-200205000-00007; doi: 10.1002/14651858.CD004842.pub2; doi: 10.1007/s00228-007-0303-7) and the recent discover as a TRPA1 non-selective antagonism (doi: 10.1111/bph.13129). We added this information in section “Discussion” (Page 9, lines 236-238)

Question 2: “I think that it would be interesting to explain in more detail in the Discussion section the relationship between TRPA1 receptors and tested oxidative parameters in the used experimental model. Also, many factors can affect the activation of TRPA1 receptors, such as changes in tissue acidity, different ions… (Srebro et al., Neuroscience. 2016). This consideration should be mentioned in the Discussion section.”

Answer:  As requested, we added in the section “Discussion” some information about TRPA1 agonism and the relationship between TRPA1 receptors and tested oxidative parameters (Page 9-10, lines 260-287).

We hope that the new version of our manuscript has been satisfactorily improved and is now suitable for publication in Pharmaceuticals.

Sincerely yours,

Professor Gabriela Trevisan

Round 2

Reviewer 1 Report

The authors have satisfactorily responded to my questions and made the necessary changes to the manuscript.

Author Response

Santa Maria, August 16th, 2021.

Reviewer #1

Reference: Manuscript ID: pharmaceuticals-1314307 (Periorbital nociception in a progressive multiple sclerosis mouse model is dependent on TRPA1 channel activation)

We agree with the reviewer's constructive suggestions for the overall improvement of the study. As requested, we have carried out English revision in our manuscript. The certificate is attached below.

 We hope that the new version of our manuscript has been satisfactorily improved and is now suitable for publication in the Pharmaceuticals.

Sincerely yours,

Professor Gabriela Trevisan
